



## 2 Providing a non-deterministic representation of spatial variability of
## 3 precipitation in the Everest region.

Judith Eeckman[a], Pierre Chevallier[a], Aaron Boone [b], Luc Neppel [a], Anneke De
Rouw [c], Francois Delclaux [a], Devesh Koirala [d]

[a]Laboratoire HydroSciences (CNRS, IRD, Universite de Montpellier) CC 57 - Universite de
Montpellier 163, rue Auguste Broussonnet 34090 Montpellier, France;
[b]CNRM UMR 3589, Meteo-France/CNRS, Toulouse, France;
[c]Institut de Recherche pour le Developpement, Universite Pierre et Marie Curie, 4 place
Jussieu, 75252 Paris cedex 5, France;
[d]Nepal Academy of Science and Technology GPO box: 3323 Khumaltar, Lalitpur, Nepal

**ABSTRACT**
This paper provides a new representation of the effect of altitude on precipitation
that represent spatial and temporal variability of precipitation in the Everest re-
gion. Exclusive observation data are used to infer a piecewise linear function for
the relation between altitude and precipitation and significant seasonal variations
are highlighted. An original ensemble approach is applied to provide non determin-
istic water budgets for middle and high mountain catchments. Physical processes
at the soil-atmosphere interface are represented through the ISBA surface scheme.
Uncertainties associated with the model parametrization are limited by the inte-
gration of in-situ measurements of soils and vegetation properties. Uncertainties
associated with representation of the orographic effect are shown to account for up
to 16% of annual total precipitation. Annual evapotranspiration is shown to rep-
resent $26\% \pm 1\%$ of annual total precipitation for the mid-altitude catchment and
$34\% \pm 3\%$ for the high-altitude catchment. Snow fall contribution is shown to be
neglectible for the mid-altitude catchment and it represents up to $44\% \pm 8\%$ of total
precipitation for the high-altitude catchment. These simulations at the local scale
enhance current knowledge of the spatial variability of hydro-climatic processes in
high- and mid-altitude mountain environments.

**KEYWORDS**
Central Himalayas; precipitation; uncertainty analysis; ISBA surface scheme

## 34 1. Introduction

The central part of the Hindu Kush Himalaya region presents tremendous heterogene-
ity, in particular in terms of topography and climatology. The terrain ranges from
the agricultural plain of Terai to the highest peaks of the world, including Mount
Everest, over a south-north transect about 150km long (FIGURE 1).

Two main climatic processes at the synoptic scale are distinguished in the Central
Himalayas (Barros *et al.* 2000, Kansakar *et al.* 2004). First, the Indian Monsoon is

Judith Eeckman judith.eeckman-poivilliers@univ-montp2.fr





formed when moist air arriving from Bay of Bengal is forced to rise and condense on the Himalayan barrier. Dhar and Rakhecha (1981) and Bookhagen and Burbank (2010) assessed that about 80% of annual precipitation over the Central Himalayas occurs between June and September. However, the timing and intensity of this summer monsoon is being reconsidered in the context of climate change (Bharati *et al.* 2016). The second main climatic process is a west flux that gets stuck in adequately oriented valleys, and occurs between January and March. Regarding high altitudes (> 3000 m), this winter precipitation can occur exclusively in solid form and can account for up to 40% of annual precipitation (Lang and Barros 2004) with considerable spatial and temporal variation.

At a large spatio temporal scale, precipitation patterns over the Himalayan range are recognized to be strongly dependent on topography (Anders *et al.* 2006, Bookhagen and Burbank 2006, Shrestha *et al.* 2012). The main thermodynamic process is an adiabatic expansion when air masses rise, but, at very high altitudes (> 4000 m), the reduction of available moisture is a concurrent process. Altitudinal thresholds of precipitation can then be discerned (Alpert 1986, Roe 2005). However, this representation of orographic precipitation has to be modulated considering the influence of such a protruding relief (Barros *et al.* 2004).

Products for precipitation estimation currently available in this area, e.g. the APHRODITE interpolation product (Yatagai *et al.* 2012) and the TRMM remote product (Bookhagen and Burbank 2006), do not represent spatial and temporal variability of orographic effects at a resolution smaller than 10 km (Gonga-Saholiariliva *et al.* 2016). Consequently, substantial uncertainty remains in water budgets simulated for this region, as highlighted by Savéan *et al.* (2015). In this context, ground-based measurements condensed in small areas have been shown to enhance the characterization of local variability of orographic processes (Andermann *et al.* 2011, Pellicciotti *et al.* 2012, Immerzeel *et al.* 2014). However, even if the Everest region is one of the most closely monitored areas of the Himalayan range, valuable observations remain scarce. In particular, the relation between altitude and precipitation is still poorly documented.

The objective of this paper is to provide a representation of the effect of altitude on precipitation that represent spatial and temporal variability of precipitation in the Everest region. The parameters controlling the shape of the altitudinal factor are constrained through an original sensitivity analysis step. Uncertainties associated with variables simulated through the ISBA surface scheme (Noilhan and Planton 1989) are quantified.

The first section of the paper presents the observation network and recorded data. The second section describes the model chosen to represent orographic precipitation, including computed altitude lapse rates for air temperature and precipitation. The method for statistical analysis through hydrological modeling is also described. The third section presents and discusses the results of sensitivity analysis and uncertainty analysis.





## 2.   Data and associated uncertainties

### 2.1.   *Meteorological station transect*

An observation network of ten stations (FIGURE 1) records hourly precipitation (P) and air temperature (T) since 2010 and 2014. The stations are equipped with classical rain gauges and HOBO ® sensors for temperature. The stations are located to depict altitudinal profile of P and T over 1) the main river valley (Dudh Koshi valley), oriented south-north; 2) the Kharikhola tributary river, oriented east-west.

To reduce under-catching of solid precipitation, two Geonors® were installed at 4218 m and 5035 m in 2013. Measurements at Geonor® instrumentation allow to correct the effect of wind and the loss of snowflakes. Records from four other stations administrated by the EVK2-CNR association are also available. Total precipitation, air temperature, atmospheric pressure (AP), relative humidity (RH), wind speed (WS), short-wave radiation (direct and diffuse) (SW) and long-wave radiation (LW) have been recorded at the hourly time step since 2000 at Pyramid station (5035 m.a.s.l.). Overall, these ten stations cover an altitude range from 2078 m to 5035 m a.s.l., comprising a highly dense observation network, compared to the scarcity of ground-based data in this type of environment.

**Table 1.** Overview of the observation network used in this study. Air temperature (T), precipitation (P) atmospheric pressure (AP), relative humidity (RH), wind speed (WS), short- and long-wave radiation (SW, LW) are recorded at the hourly time scale. The Geonor® at the Pyramid and Pheriche stations record total precipitation $P_{GEO}$ at the hourly time scale. The two hydrometric stations at Kharikhola and Pangboche record water level since 2014.

| ID | Station | ALT m.a.s.l. | LAT | LON | Period | | Measured variable |
|---|---|---|---|---|---|---|---|
| KHA | Kharikhola | 2078 | 27.60292 | 86.70311 | 2014-05-03 | 2015-10-28 | P,T |
| MER | Mera School | 2561 | 27.60000 | 86.72269 | 2014-05-02 | 2015-10-28 | P,T |
| BAL | Bhalukhop | 2575 | 27.60097 | 86.74017 | 2014-05-03 | 2015-10-28 | P,T |
| PHA | Phakding | 2619 | 27.74661 | 86.71300 | 2010-04-07 | 2016-05-16 | P,T |
| LUK | Lukla | 2860 | 27.69694 | 86.72270 | 2002-11-02 | 2016-01-01 | P,T |
| PAR | Paramdingma | 2869 | 27.58492 | 86.73956 | 2014-05-03 | 2015-10-28 | P,T |
| TCM | Pangom | 3022 | 27.58803 | 86.74828 | 2014-05-03 | 2015-10-28 | P,T |
| NAM | Namche | 3570 | 27.80250 | 86.71445 | 2001-10-27 | 2016-01-01 | P,T |
| PAN | Pangboche | 3976 | 27.85722 | 86.79417 | 2010-10-29 | 2016-05-08 | P,T |
| PHE | Pheriche | 4218 | 27.89528 | 86.81889 | 2001-10-25 | 2016-01-01 | T |
| | | | | | 2012-12-06 | 2016-05-16 | $P_{GEO}$ |
| PYR | Pyramid | 5035 | 27.95917 | 86.81333 | 2000-10-01 | 2016-01-01 | T,AP,RH,WS, LW,SW |
| | | | | | 2016-04-26 | 2016-04-26 | $P_{GEO}$ |
| 668.7 | Kharikhola | 1985 | 27.60660 | 86.71847 | 2014-05-03 | 2016-05-20 | Water level |
| 668.03 | Pangboche | 3976 | 27.85858 | 86.79253 | 2014-05-17 | 2016-05-09 | Water level |

Annual means for temperature and precipitation measured at these stations are presented are presented TABLE 2 for the two hydrological years 2014-2015 and 2015-2016. These time series present up to 61% missing values. For stations LUK, NAM, PHA, PAN, PHE and PYR, where relatively long time series are available, gaps were filled with the interannual hourly mean for each variable. For the other stations, gaps were filled with values at the closest station, weighted by the ratio of mean values over the commun periods. Time series from 2013-01-01 to 2016-04-30 were then reconstructed from these observations.





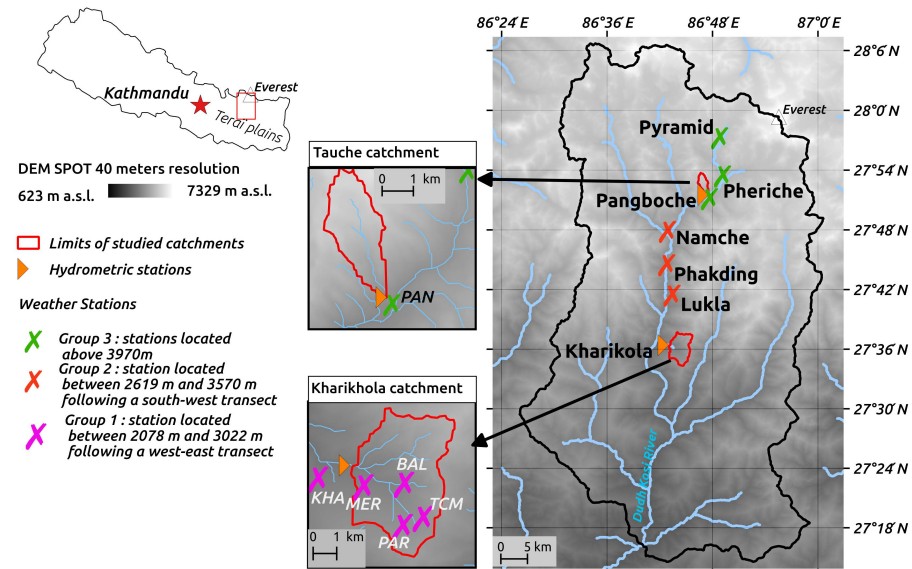

**Figure 1.** Map of the monitored area: the Dudh Koshi River basin at the Rabuwabazar station, managed by the Department of Hydrology and Meteorology, Nepal Government (station coordinates: $27°16'09''N$, $86°40'03''E$, station elevation: 462 m a.s.l., basin area: 3712 km2). The Tauche and Kharikhola subcatchments are defined by the corresponding limnimetric stations.

Two seasons are defined based on these observations and knowledge of the climatology of the Central Himalayas: 1) the monsoon season, from April to September, including the early monsoon, whose influence seems to be increasing with the recent climate change (Bharati *et al.* 2014); 2) the winter season, dominated by westerly entrances with a substantial spatiotemporal variability.

Local measurements are necessarily biased by aleatory errors (according to Beven (2015) uncertainty classification). In particular, snowfall is usually undercaught by instrumentation (Sevruk *et al.* 2009). However, since this study focuses most particularly on uncertainty associated with spatialization of local measurements, aleatory errors in measurements will not be considered here.

## 2.2. Discharge measurement stations and associated hydrological catchments

Two hydrometric stations were equipped with Campell® hydrometric sensors and encompass two sub-basins: Kharikhola catchment ($18.2 km^2$) covers altitudes from 1900 m to 4450 m (mid-altitude mountain catchment) and Tauche catchment ($4.65 km^2$) altitudes range from 3700 m to 6400 m (high-altitude mountain catchment). Water level time series are available from March 2014 to March 2015, with 16.5% and 0% data gaps, respectively (TABLE 3). Uncertainty on discharge is usually considered to account for less than 15% of discharge (Lang *et al.* 2006).





Recession times are computed on available recession periods using the lfstat R li-
brary (Koffler and Laaha 2013) with both the recession curves method (World Mete-
orological Organization 2008) and the base flow index method (Chapman 1999). We
found recession times for Kharikhola and Tauche catchment of respectively around
70 days and around 67 days. Consequently, we consider that there is no interannual
storage in either of the two catchments. This hypothesis can be modulated if a con-
tribution of deep groundwater is considered (Andermann *et al.* 2011). Since these two
catchments have null (Kharikhola) or neglectible (Tauche) glacier contribution, we
hypothesized that the only entrance for water budgets in these catchments is total
precipitation. In this study we used these two catchments as samples to assess gener-
ated precipitation fields against observed discharge at the local scale. The hydrological
147 year is considered to start on 1 April, as decided by the Department of Hydrology and
148 Meteorology of the Nepalese Government and generally considered (Nepal *et al.* 2014,
Savéan *et al.* 2015).

**Table 2.** Overview of measurements at meteorological stations used in this study over the hydrological years 2014-2015 and 2015-2016. $\overline{T}$, $\overline{P}$ stand for, respectively, annual mean temperatur and annual total precipitation. $\overline{T}$, $\overline{P}$ are computed on time series completed with either a weighted value at the closest station when available, or their respective interannual mean.

| | 2014-2015 | | | | 2015-2016 | | | |
| | Temperature | | Precipitation | | Temperature | | Precipitation | |
| Station | $\overline{T}$ | Gaps | $\overline{P}$ | Gaps | $\overline{T}$ | Gaps | $\overline{P}$ | Gaps |
| | °C | | mm | | °C | | mm | |
|---|---|---|---|---|---|---|---|---|
| KHA | 13.96 | 0.1% | 2453 | 34.5% | 15.50 | 100% | 1752 | 100% |
| MER | 13.44 | 12.4% | 3241 | 12.2% | 14.83 | 100% | 2278 | 100% |
| BAL | 9.92 | 15.1% | 3679 | 34.4% | 10.48 | 0.0% | 2628 | 0.0% |
| PHA | 9.26 | 41.9% | 1664 | 0.0% | 9.16 | 0.0% | 1226 | 0.0% |
| LUK | 10.18 | 54.5% | 2278 | 41.8% | 10.19 | 40% | 2278 | 0.2% |
| PAR | 7.98 | 20% | 3592 | 19.8% | 7.84 | 100% | 2540 | 100% |
| TCM | 7.07 | 21.1% | 3592 | 20.8% | 6.90 | 100% | 2628 | 100% |
| NAM | 5.09 | 19.9% | 964 | 0.1% | 5.17 | 57.9% | 788 | 0.1% |
| PAN | 3.81 | 0.2% | 876 | 0.0% | 4.20 | 0.0% | 526 | 0.0% |
| PHE | 0.80 | 61% | 701 | 0.0% | 0.84 | 8.6% | 526 | 0.0% |
| PYR | -2.71 | 18.6% | 701 | 0.0% | -2.30 | 9.3% | 438 | 0.0% |

**Table 3.** Overview of measurements at hydrological stations used in this study over the hydrological years 2014-2015 and 2015-2016. $\overline{Q}$ stands for annual discharge. $\overline{Q}$ for the Kharikhola station in 2014-2015 is completed with the interannual mean.

| | 2014-2015 | | 2015-2016 | |
| Station | $\overline{Q}$ | Gaps | $\overline{Q}$ | Gaps |
| | mm | | mm | |
|---|---|---|---|---|
| Kharikhola | 2341 | 34.0% | 1746 | 0.0% |
| Pangboche | 416 | 0.0% | 499 | 0.0% |





## 3. Spatialization methods for temperature and precipitation

### 3.1. *Temperature*

In mountainous areas, temperature and altitude generally correlate well linearly, considering a large time scale (Valéry *et al.* 2010, Gottardi *et al.* 2012). In the majority of studies based on field observations, air temperature values are extrapolated using the inverse distance weighting method (IDW) (Andermann *et al.* 2012, Immerzeel *et al.* 2012, Nepal *et al.* 2014). An altitude lapse rate $\theta$ (in $°C.km^{-1}$) is also used to take altitude into account for hourly temperature computation at any point M of the mesh extrapolated by IDW (EQUATION 1).

$$T(M) = \frac{\displaystyle\sum_{S_i} d^{-1}(M, S_i).(T(S_i) + \theta.(z_m - z_i))}{\displaystyle\sum_{S_i} d^{-1}(M, S_i)} \qquad (1)$$

where $T$ is the hourly temperature, $S_i$ the $i$th station of the observation network, $z_i$ the altitude of station $S_i$, $z_M$ altitude of grid point M and $d^{-1}$ is the inverse of distance in latitude and longitude.

In the Himalayas, seasonal (Nepal *et al.* 2014, Ragettli *et al.* 2015) or constant (Pokhrel *et al.* 2014) altitudinal lapse rates (LR) are used for temperature. FIGURE 2 presents seasonal LR computed from temperature time series at the 10 stations described in section 2.1. The linearity is particularly satisfying for both seasons, even if stations follow differently oriented transects (W-E or N-S orientation). Computed LR for both seasons are very close to values proposed by Immerzeel *et al.* (2014), Heynen *et al.* (2016) (Langtang catchment, 585 $km^2$, elevation ranging from 1406 m.a.s.l. to 7234 m.a.s.l.) and Salerno *et al.* (2015) (Koshi basin, $58100 km^2$, from 77 m.a.s.l. to 8848 m.a.s.l.). Consequently, these values for seasonal LR will be used in this study. Uncertainties associated with temperature interpolation will therefore be neglected, because they have minor impact on modelling compared to uncertainties on precipitation.

### 3.2. *Precipitation*

#### 3.2.1. *Model of orographic precipitation*

The complexity of precipitation spatialization methods has beend commented by Barros and Lettenmaier (1993). When orographic effects are not well understood, complex approaches do not necessarily reproduce local measurements efficiently (Bénichou and Le Breton 1987, Frei and Schär 1998, Daly *et al.* 2002). In the Central Himalayas, various hydrologic and glaciological studies are based on observation networks to produce a precipitation grid. They use either observed altitude lapse rates, e.g., in the Langtang range, (Immerzeel *et al.* 2012, Ragettli *et al.* 2015) and in the Dudh Koshi River basin, (Nepal *et al.* 2014), or geostatistical methods (Gonga-Saholiariliva *et al.* 2016) (Koshi catchment). Nevertheless, the IDW method is a simple, widely used method to spatialize precipitation. In the French Alps, Valéry *et al.* (2010) combine the IDW method with a multiplicative altitudinal factor. Precipitation at any point M of the mesh extrapolated by the IDW is given by EQUATION 2.





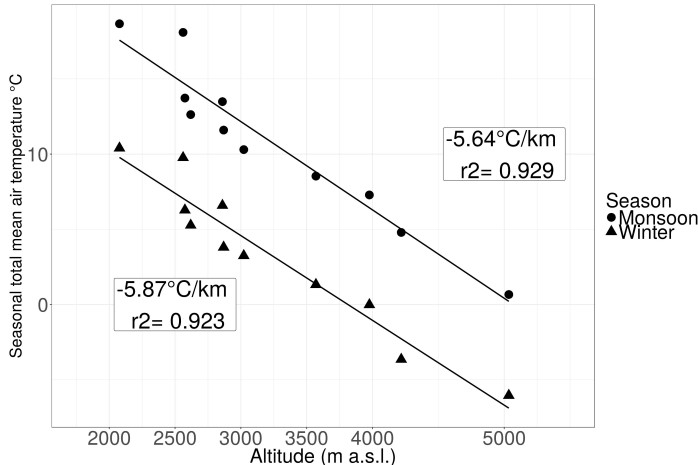

**Figure 2.** Linear regression for measured seasonal temperatures for the winter and monsoon seasons. Points (circles or triangles) are the seasonal means at each monitored station. Altitude lapse rates are displayed for each season in $°C.km^{-1}$.

$$P(M) = \frac{\sum_{S_i} d^{-1}(M, S_i).(P(S_i).exp(\beta(z_M - z_i)))}{\sum_{S_i} d^{-1}(M, S_i)} \qquad (2)$$

In EQUATION 2, the altitude effect is represented through the introduction of the
altitudinal factor $\beta$, defined by Valéry *et al.* (2010) as the slope of the linear regression
between the altitude of stations (in m.a.s.l.) and the logarithm of seasonal volume of
total precipitation expressed in millimeters.

*3.2.2.  Observed relation between altitude and seasonal precipitation*

Several studies based on observations (Dhar and Rakhecha 1981, Barros *et al.*
2000, Bookhagen and Burbank 2006, Immerzeel *et al.* 2014, Salerno *et al.* 2015),
or theoretical approaches (Burns 1953, Alpert 1986) observed that precipitation in
the Himalayan range generally presents a multimodal distribution along elevation.
Precipitation is considered to increase with altitude until a first altitudinal threshold
located between 1800 m and 2500 m, depending on the study, and to decrease above
2500m. Moreover, the linear correlation of precipitation with altitude is reported to
be weak for measurements above 4000 m (Salerno *et al.* 2015). The descreasing of
precipitation with altitude are characterized through various fonctions (Dhar and
Rakhecha 1981, Bookhagen and Burbank 2006, Salerno *et al.* 2015). Nevertheless,
the hypothesis of linearity of precipitation (P) with altitude (z) is often made, with
a constant (Nepal *et al.* 2014) or time-dependent lapse rate (Immerzeel *et al.* 2014).
Gottardi *et al.* (2012) noted that, in mountainous areas, the hypothesis of a linear
relation between P and z is only acceptable over a small spatial extension and for
homogeneous weather types. Consequently, we considered altitude lapse rates for





precipitation at the seasonal time scale, and we analyzed the spatial variability of the relation between P and z.

For this purpose, we chose to regroup the stations into three groups (see FIGURE 1): 1) stations with elevation ranging from 2078 m to 3022 m, following a west-east transect (Group 1) ; 2) stations with elevation ranging from 2619 m to 3570 m following a south-west transect (Group 2); and 3) stations with elevation above 3970 m (Group 3). FIGURE 3 shows that 1) for Group 1, observed seasonal volumes of precipitation increase globally with altitude at a rate lower than $0.1 km^{-1}$; 2) for Group 2, seasonal volumes decrease at a rate around $-0.3 km^{-1}$ ; 3) for Group 3, seasonal volumes decrease at a rate lower than $0.2 km^{-1}$, with a poor linear trend.

The overlapping of altitude ranges between Group 1 and Group 2 highlights that the relation between precipitation and altitude strongly depends on terrain orientation. The difference in seasonal volumes at the BAL (2575 m a.s.l., 3471 mm/year) and MER stations (2561 m a.s.l., 2245 mm/year) (GROUP 1) also result from site effects on precipitation. In summary, $\beta$ values inferred from local observations mainly express local variability and are not sufficient to establish any explicit relation between precipitation and altitude at the catchment scale. However, for operational purposes, the $\beta$ factor can be simplified as a multi-modal function of altitude within the Dudh Koshi catchment. Optimum values that optimally fit local variability were then investigated through a sensitivity and uncertainty analysis. The $\beta$ factor is represented as a piecewise linear function of altitude using two altitudinal thresholds $z_1$ and $z_2$ and three altitude lapse rates $\beta_1$, $\beta_2$ and $\beta_3$ (EQUATION 3).

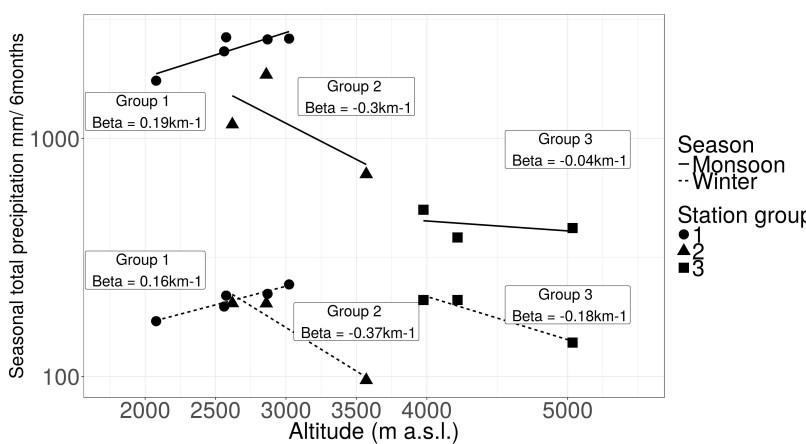

**Figure 3.** Piecewise relation between altitude and the logarithm of observed seasonal volumes of total precipitation, separated by season and station group. Seasonal values for $\beta$ $(km^{-1})$ are computed from observed precipitation for each of the three station groups.

$$\beta(z) = \begin{cases} \beta_1 > 0 & \text{if} & z \leq z_1 \\ \beta_2 < 0 & \text{if} & z_1 < z \leq z_2 \\ \beta_3 \sim 0 & \text{if} & z > z_2 \end{cases} \quad (3)$$





Since no deterministic value can be ensured for the five parameters controlling the shape of EQUATION 3, an ensemble approach was applied (see Section 4) to estimate parameter sets at the scale of the entire Dudh Koshi River basin that are optimally suitable for both Tauche and Kharikhola catchments.

## 4. Sensitivity and uncertainties analysis method

### 4.1. *Overall strategy*

Saltelli *et al.* (2006) distinguishes sensitivity analysis (SA), which does not provide a measurement of error, and uncertainties analysis (UA), which computes a likelihood function according to reference data. SA is run before UA as a diagnostic tool, in particular to reduce variation intervals for parameters and therefore save computation time.

The algorithm chosen for SA was the Regional Sensitivity Analysis (RSA) (Spear and Hornberger 1980) method. The RSA method is based on the separation of the parameter space into (at least) two groups: behavioral or nonbehavioral parameter sets. A behavioral parameter set is a set that respects conditions (maximum or minimum thresholds) on the output of the orographic precipitation model. Thresholds will be defined for solid and total precipitation in the Results section. The analysis is performed using the R version of the SAFE(R) toolbox, developed by Pianosi *et al.* (2015).

SA and UA are set up as follows (Beven 2010):

(1) First, the parameter space is sampled, according to a given sampling distribution. For each parameter set, hourly precipitation fields are computed at the 1-km resolution using EQUATION 2 for both the Tauche and Kharikhola catchments. Since physical processes condition the relation between altitude and precipitation strongly differ between the two seasons, we chose to distinguish altitude correction for the winter and monsoon seasons. Behavioral parameter sets were then selected for each of the two seasons.

(2) Then, for each behavioral precipitation field, the ISBA surface scheme, described in the next section, was run separately on Kharikhola and Tauche catchments. The objective function was computed as the difference between simulated and observed annual discharge at the outlet of each catchment. Parameter sets that lead to acceptable discharge regarding observed discharge for the two catchments are finally selected.

### 4.2. *Hydrological modeling at the local scale*

#### 4.2.1. *The ISBA surface scheme*

We considered that there was no interannual storage in either of the two subcatchments studied, i.e., the variation of the groundwater content was considered null from one hydrological year to the other. Consequently, annual simulated discharges were computed as the sum over all grid cells and all time steps, of simulated surface flow and simulated subsurface flow. The question of calibration of flow routing in the catchment was thus avoided.





The ISBA surface scheme (Noilhan and Planton 1989, Noilhan and Mahfouf 1996) simulates interactions between the soil, vegetation and the atmosphere on a sub-hourly time step (SVAT model). The multi-layer version of ISBA (ISBA-DIF) uses a diffusive approach (Boone *et al.* 2000, Decharme *et al.* 2011): surface and soil water fluxes are propagated from the surface through the soil column. Transport equations for mass and energy are solving using a multilayer vertical discretization of the soil. The explicit snow scheme in ISBA (ISBA-ES) Boone and Etchevers (2001) uses a three-layer vertical discretization of snow pack and provides a mass and energy balance for each layer (Boone and Etchevers 2001). Snow-melt and snow sublimation are taken into account in balance equations. The separation between runoff over saturated areas (Dunne runoff), infiltration excess runoff (Horton runoff) and infiltration is controled by the Variable Infiltration Capacity Scheme (VIC) (Dümenil and Todini 1992).

The precipitation phase was estimated depending on hourly air temperature readings. Mixed phases occurred for temperatures between 0°C and 2°C, following a linear relation. Other input variables required for ISBA (atmospheric pressure, relative humidity, wind speed, short- and long-wave radiations) are interpolated from measurements at Pyramid station as functions of altitude, using the method proposed by Cosgrove *et al.* (2003). Short wave radiation and wind speed are not spatially interpolated and are considered to be equals to the measurements at Pyramid station for the two catchments.

### 4.2.2. Parametrization of surfaces

Several products provide parameter sets for physical properties of surfaces at the global scale (Hagemann 2002, Masson *et al.* 2003, Arino *et al.* 2012). However, these products are not accurate enough at the resolution required for this study. The most recent analysis (Bharati *et al.* 2014, Ragettli *et al.* 2015) exclusively used knowledge garnered from the literature. To detail the approach, in this study the parametrization was based on in situ measurements. A classification into nine classes of soil/vegetation entities was defined based on Sentinel2 images at a 10-m resolution (Drusch *et al.* 2012), using a supervised classification tool of the QGIS Semi-Automatic Classification Plugin (Congedo 2015).

In and around the two cathchments, 24 reference sites were sampled during field missions. Data collection included soil texture, soil depth, root depth, determined by augering to a maximum depth of 1.2m. Vegetation height, structure and dominant plant species were also determined. The results were classified into nine surface types. The nine classes and their respective fractions in Kharikhola and Tauche catchments are presented TABLE 4.

Analysis of soil samples showed that soils were mostly sandy ($\sim$ 70%), with a small proportion of clay ($\sim$ 1%). Soil depths varied from very thin ($\sim$ 30 cm) at high altitudes to 1.2 m for flat cultivated areas. Forest areas were separated into three classes: dry forests were characterized by high slopes and shallow soils; wet forests presented deep silty soils (1 m), with high trees (7 m). Intermediate forests had moderate slopes and relatively deep, sandy soils. Crop areas presented different soil depths depending on their average slope. In addition, values for unmeasured variables (LAI, soil and vegetation albedos, surface emissivity, surface roughness) were taken from the ECO-CLIMAP classification (Masson *et al.* 2003) for ecosystems representative of the study





area. ECOCLIMAP provides the annual cycle of dynamic vegetation variables, based
both on a surface properties classification (Hagemann 2002) and on a global climate
map (Koeppe and De Long 1958).

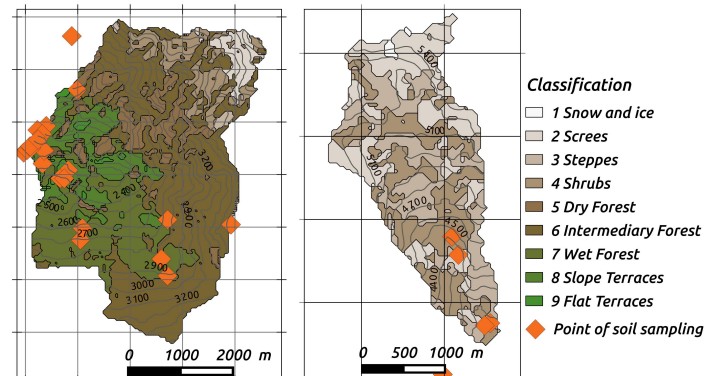

**Figure 4.** Classification of surfaces defined for the two Kharikhola and Tauche subcatchments, established using the supervised classification tool of the QGIS Semi-Automatic Classification Plugin (Congedo 2015), based on Sentinel2 images at a 10-m resolution (Drusch *et al.* 2012). In situ sample points were used to describe the soil and vegetation characteristics of each class.

**Table 4.** Soil and vegetation characteristics of the nine classes defined in Kharikhola and Tauche catchments, respectively. % KK and % Tauche are the fraction of each class on Kharikhola and Tauche catchments. Sand and clay fractions (% Sand and % Clay, respectively), soil depth (SD), root depth (RD) and tree height (TH) are defined based on in situ measurements. The dynamic variables (e.g. the fraction of vegetation and Leaf Area Index) were found in the ECOCLIMAP classification (Masson *et al.* 2003) for representative ecosystems.

| ID | Class | % KK | % Tauche | % Sand | % Clay | TH m | SD m | RD m | Ecoclimap Cover |
|----|-------|------|----------|--------|--------|------|------|------|-----------------|
| 1 | Snow and ice | - | 0.7% | 0.00 | 0.00 | 0.0 | 0.00 | 0.00 | 6 |
| 2 | Screes | 3.1% | 31.2% | 0.00 | 0.00 | 0.0 | 0.00 | 0.00 | 5 |
| 3 | Steppe | 0.6% | 33.7% | 81.41 | 1.70 | 0.0 | 0.10 | 0.10 | 123 |
| 4 | Shrubs | 7.4% | 34.4% | 70.60 | 1.55 | 0.0 | 0.35 | 0.27 | 86 |
| 5 | Dry Forest | 9.7% | - | 72.86 | 1.00 | 12.0 | 0.20 | 0.20 | 27 |
| 6 | Intermediary Forest | 45.7% | - | 84.97 | 1.01 | 27.5 | 0.42 | 0.40 | 27 |
| 7 | Wet Forest | 20.6% | - | 70.12 | 1.00 | 6.8 | 1.04 | 0.50 | 27 |
| 8 | Slope terraces | 11.2% | - | 70.89 | 1.38 | 5.6 | 0.56 | 0.26 | 171 |
| 9 | Flat terraces | 1.4% | - | 67.01 | 1.69 | 2.5 | 1.267 | 0.20 | 171 |

## 5. Results and discussion

### 5.1. *Regional sensitivity analysis*

The parameter space was sampled using the All at a time (AAT) sampling algorithm
from the SAFE(R) toolbox (Pianosi *et al.* 2015). Since no particular information was
available on prior distribution and interaction for the five parameters, uniform distri-
butions were considered. The initial ranges for $\beta_1$, $\beta_2$ and $\beta_3$ parameters were defined
based on the lapse rates computed at the seasonal time scale from observations. Ranges





for altitudinal thresholds z1 and z2 were deduced from other studies (Bookhagen and Burbank 2006, Nepal 2012, Savéan 2014). The initial ranges are given in TABLE 6. The size of parameter samples was chosen according to Sarrazin *et al.* (2016) (TABLE 5) .

Maximum and minimum conditions on annual total precipitation for a set to be behavioral were chosen according to annual observed discharge for each of the two catchments. The mean observed discharge for the recorded period was 2043 mm/year at the Kharikhola station and 457 mm/year at the Tauche station. Annual total precipitation was expected to be greater than the measured annual discharge and lower than annual discharge plus 70%. These thresholds take into account both the uncertainty on measured discharges and actual evapotranspiration. Based no a values proposed in the literature, evapotranspiration is assumed to represent less than 50% of observed discharge, for both catchments. The minimum and maximum thresholds for both catchments are summarized TABLE 7.

The method's convergence (i.e., the stability of the result when the sample size grows) was graphically assessed. The results converged for sample sizes from 1000 samples. FIGURE 5 shows the cumulative density function (CDF) for behavioral and nonbehavioral parameter sets for the monsoon and winter seasons. Of the 2000 parameter sets sampled, 712 sets verified the chosen minimum and maximum conditions for annual total precipitation and snowfall (i.e., they were behavioral). The sensitivity of the output to each parameter was evaluated by the maximum vertical distance (MVD) between CDF for behavioral and nonbehavioral parameter sets. Annual total precipitation appeared to be less sensitive to parameters controlling winter precipitation than to parameters controlling monsoon precipitation. This result can be explained by the fact that winter precipitation was less than monsoon precipitation. However, since the applied sampling method does not take into account the existing interaction between the five parameters, further analysis for parameter ranking was not significant.

The method was necessarily sensitive to the prior hypothesis presented TABLE 5. In particular, the conditions for a set to be behavioral have a significant impact on the distribution of the behavioral sets. On the contrary, increasing the sample size does not affect the output distribution, since minimum size for convergence is reached.

**Table 5.** The algorithm selected, sample size and prior distribution for sampling the parameter space using the SAFE(R) toolbox (Pianosi *et al.* 2015).

| | |
|---|---|
| Sample size | 2000 |
| Nb. of model evaluation | 2000 |
| Sampling algorithm | All-at-a-Time |
| Sampling method | Latin Hypercube |
| Prior distributions | Uniforms |





**Table 6.** Initial ranges considered for the five shape parameters of the altitudinal factor: z1, z2 $\beta_1$, $\beta_2$ and $\beta_3$. Ranges are defined based on measurements at stations and on values founded in the literature.

|          | Minimum | Maximum |            |
|----------|---------|---------|------------|
| z1       | 1900    | 3500    | m. a.s.l.  |
| z2       | 3500    | 6500    | m. a.s.l.  |
| $\beta_1$ | 0.00   | 2.00    | $km^{-1}$  |
| $\beta_2$ | -2.00  | 0.00    | $km^{-1}$  |
| $\beta_3$ | -0.30  | 0.00    | $km^{-1}$  |

**Table 7.** Conditions over total precipitation on the Kharikhola and Tauche catchments for a parameter set to be behavioral. Annual total precipitation was expected to be greater than the measured annual discharge plus 20% and lower than annual discharge plus 50%.

|           | Minimum | Maximum |          |
|-----------|---------|---------|----------|
| Kharikhola | 2043   | 3473    | mm/year  |
| Tauche    | 457     | 777     | mm/year  |

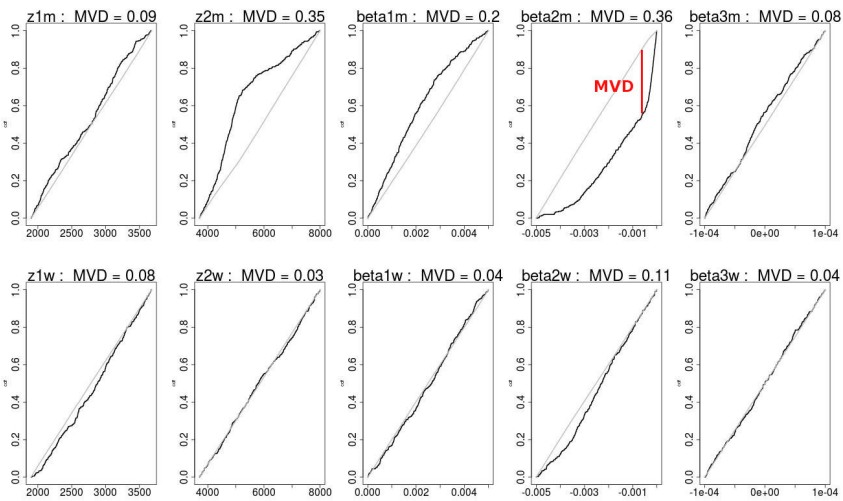

**Figure 5.** Cumulative density function of behavioral and non-behavioral output for each parameter for the two seasons. Black lines are cumulative distributions of behavioral parameter sets, and grey lines are cumulative distributions of non-behavioral sets. Parameters with indication w (respectively, m) stand for winter values (respectively, monsoon values). The greater the maximum vertical distance (MVD), the more influential the parameter was. MVD is drawn as an example for parameter beta2m.

## 5.2. *Uncertainties analysis*

### 5.2.1. *Annual simulated water budgets*

The precipitation fields generated using each behavioral parameter set were used as input data within the ISBA surface scheme. The simulation over the Tauche and Kharikhola catchments were run separately over the 2013-01-01/2016-03-31 period, at the hourly time scale. The 2013–2014 year was used as a spin-up period and the results were observed for 2014–2015 and 2015–2016 hydrological years. To overcome the issue





**Table 8.** Mean values (X), standard deviation ($\overline{X}$) and relative standard deviation ($\sigma/\overline{X}$) for total precipitation (PTOT), snowfall (SNOWF), discharge (RUNOFF) and actual evapotranspiration (EVAP) simulated with ISBA for the Kharikhola catchment or Tauche catchment: the mean for 2014–2015 and 2015–2016.

| | Kharikhola catchment | | | | | | Tauche catchment | | | | | |
| | 2014-2015 | | | 2015-2016 | | | 2014-2015 | | | 2015-2016 | | |
| | $\overline{X}$ | $\sigma$ | $\sigma/\overline{X}$ | $\overline{X}$ | $\sigma$ | $\sigma/\overline{X}$ | $\overline{X}$ | $\sigma$ | $\sigma/\overline{X}$ | $\overline{X}$ | $\sigma$ | $\sigma/\overline{X}$ |
| | mm | mm | - | mm | mm | - | mm | mm | - | mm | mm | - |
| EVAP | 604 | 17 | 3% | 664 | 16 | 2% | 213 | 16 | 8% | 219 | 15 | 7% |
| PTOT | 2868 | 295 | 10% | 2069 | 207 | 10% | 766 | 110 | 14% | 525 | 82 | 16% |
| RUNOFF | 2279 | 293 | 13% | 1421 | 203 | 14% | 517 | 128 | 25% | 459 | 85 | 19% |
| SNOWF | 32 | 8 | 25% | 22 | 7 | 32% | 364 | 56 | 15% | 205 | 35 | 17% |

of calibrating a flow-routing module, the simulated discharge were aggregated at the annual time scale and compared to annual observed discharge at the outlet ($\overline{Q}_{obs}$).

FIGURE 6 presents boxplots obtained for the 712 behavioral parameter sets for the terms of the annual water budget, i.e., liquid and solid precipitation, discharge and evapotranspiration. The dotted line represents $\overline{Q}_{obs}$ for each catchment. The mean annual volumes of simulated variables were also computed for each parameter set in 2014–2015 and 2015–2016, and the intervals of uncertainty associated with simulated annual volumes are provided. This method highlights the propagation of uncertainties associated with the representation of orographic effects toward simulated terms of annual water budgets.

TABLE 8 presents the mean value, standard deviation and relative standard deviation for all of the ISBA simulated variables for the Kharikhola and Tauche catchments, for 2014–2015 and 2015–2016. The annual actual evapotranspiration accounted for 26% of annual total precipitation for Kharikhola and 34% for Tauche. In comparison, evapotranspiration was estimated at about 20%, 14% and 53% of total annual precipitation, respectively, by (Andermann *et al.* 2012),(Nepal *et al.* 2014) and (Savéan *et al.* 2015) over the entire Dudh Koshi basin and (Ragettli *et al.* 2015) estimated it at 36.2% of annual total precipitation for the upper part of the Langtang basin.

Annual snow fall volume for Kharikhola was a neglectible fraction of annual total precipitation (∼1%) and it was around 44% for Tauche. Annual snowfall was estimated at, respectively, 15.6% and 51.4% of annual total precipitation by (Savéan *et al.* 2015) (entire Dudh Koshi river basin) and (Ragettli *et al.* 2015) (upper part of the Langtang basin).

Moreover, this statistical approach shows that the only uncertainties associated with representation of the orographic effect results in significant uncertainties on simulated variables. These uncertainties account for up to 16% for annual total precipitation, up to 25% for annual discharge and up to 8% for annual actual evapotranspiration. Uncertainty on annual snowfall is quantified at 16% for high mountain catchment and up to 32% for middle mountain catchment. These uncertainty intervals are essentially conditioned by model structure and parametrization, and these results point out that simulated water budgets provided by modelling studies must necessarily be associated with error intervals.



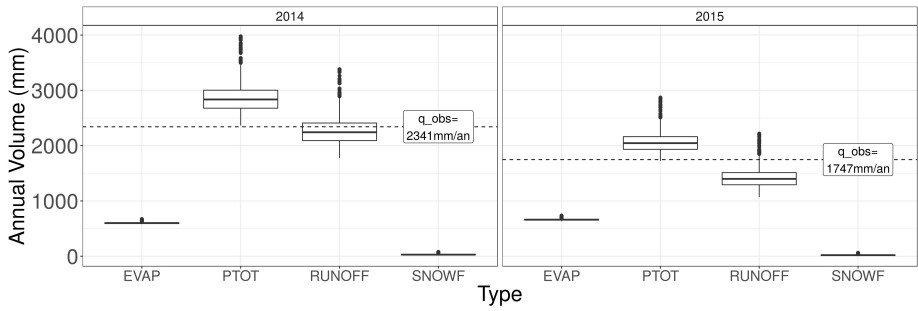

(a) For Kharikhola catchment.

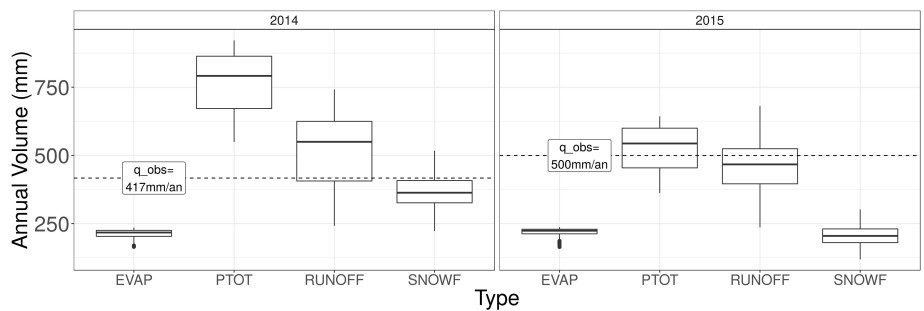

(b) For Tauche catchment.

**Figure 6.** Boxplots for distribution of annual volumes of the terms of the water budget: Discharge (RUNOFF), solid and total precipitation (SNOWF and PTOT) and evapotranspiration (EVAP) for 2014–2015, for the Kharikhola and Tauche catchments.

### 5.2.2. Toward optimizing parameter sets with bias on annual discharge

Going further into the simulation results, the hydrological cycle was inverted, in order to use observed discharge to optimize the relation between precipitation and altitude, as presented for mountainous areas by Valéry *et al.* (2009). Precipitation fields were then constrained at the local scale according to simulated discharges. Annual bias on discharge were computed for each catchment as the absolute value of the ratio between the observed and simulated annual discharges minus 1. FIGURE 7 presents the scatter plot of the distributions of bias on annual discharge for the Kharikhola and Tauche catchments. The Pareto optimums minimizing bias on annual discharge for both catchments were computed using the R rPref package (Roocks and Roocks 2016). For exemple, the ten first Pareto optimums were selected among the 712 behavioral parameter sets considered. The values of parameters for the winter and monsoon seasons for the ten first optimum sets are summarized in TABLE 9. For the ten parameter sets selected, the altitudinal threshold z1 was located between 2010 m.a.s.l. and 3470 m.a.s.l. during the monsoon season and between 2287 m.a.s.l. and 3488 m.a.s.l. during winter. The second altitudinal threshold z2 was located between 3709 m.a.s.l. and 6167 m.a.s.l. during monsoon and between 3734 m.a.s.l. and 6466 m.a.s.l. during winter. Altitudes found for z1 were globally higher than altitudes proposed in the literature for the second mode of precipitation (between 1800 m.a.s.l. and 2400 m.a.s.l.,





as described in section 3.2.2). Since these values were calibrated at the local scale, ac-
cording to ground-based measurements, they can be considered to accurately represent
the local variability encountered in the Tauche and Kharikhola catchments. Moreover,
values for an altitudinal threshold of precipitation located above 4000 m.a.s.l. were
proposed.

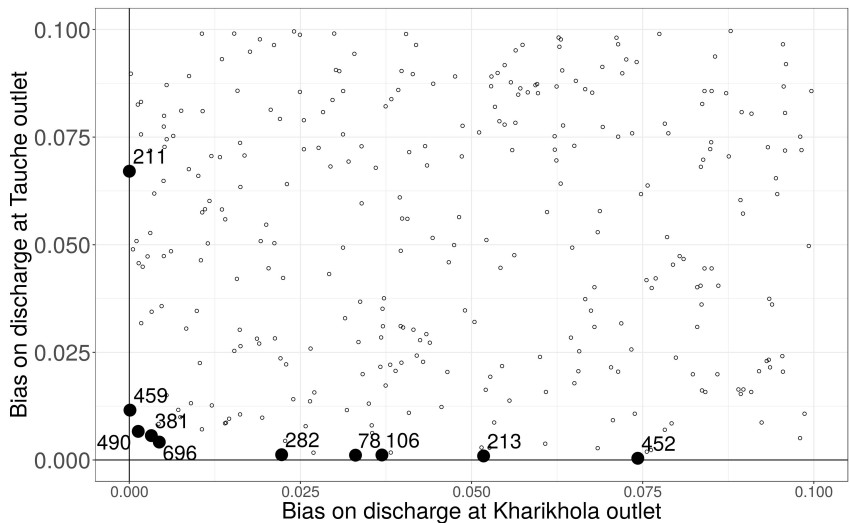

**Figure 7.** Scatter plot of bias on mean annual discharges for the Kharikhola and Tauche catchments for 2014–2015. Darker dots are parameter sets that provide the ten first Pareto optimums according to both criteria: bias for discharges on the Kharikhola and Tauche catchments. Optimal value for bias is 0. Graphical window is limited.

**Table 9.** Values of parameters for the winter and monsoon seasons for the ten first Pareto optimum sets. The Pareto optimums minimize bias on annual discharge for both catchments.

| Sample $n°$ | 78 | 106 | 211 | 213 | 282 | 381 | 452 | 459 | 490 | 696 | |
|---|---|---|---|---|---|---|---|---|---|---|---|
| z1m | 3470 | 3066 | 3286 | 2010 | 2971 | 2946 | 3337 | 2333 | 2064 | 2253 | m.a.s.l. |
| z2m | 3709 | 4938 | 6101 | 4379 | 4813 | 5596 | 5681 | 3915 | 6167 | 5978 | m.a.s.l. |
| beta1m | 0.032 | 0.028 | 0.455 | 1.772 | 1.089 | 1.755 | 0.787 | 0.73 | 0.135 | 0.003 | $km^{-1}$ |
| beta2m | -1.382 | -0.48 | -0.556 | -0.143 | -0.169 | -0.397 | -0.516 | -1.394 | -0.587 | -0.341 | $km^{-1}$ |
| beta3m | -0.283 | -0.229 | -0.059 | -0.207 | -0.298 | -0.037 | -0.003 | -0.25 | -0.033 | -0.111 | $km^{-1}$ |
| | | | | | | | | | | | |
| z1w | 3113 | 2727 | 2287 | 2895 | 3236 | 2623 | 2446 | 3488 | 2554 | 2639 | m.a.s.l. |
| z2w | 4943 | 4716 | 3871 | 6466 | 5657 | 3734 | 4336 | 5163 | 4732 | 5155 | m.a.s.l. |
| beta1w | 1.917 | 0.288 | 0.869 | 1.533 | 1.658 | 0.293 | 0.115 | 1.729 | 1.256 | 0.348 | $km^{-1}$ |
| beta2w | -1.83 | -1.096 | -1.588 | -1.791 | -0.804 | -0.455 | -1.568 | -1.457 | -1.612 | -0.508 | $km^{-1}$ |
| beta3w | -0.191 | -0.2 | -0.255 | -0.244 | -0.068 | -0.165 | -0.294 | -0.011 | -0.039 | -0.037 | $km^{-1}$ |
| bias Khari. | 0.033 | 0.037 | 0 | 0.052 | 0.022 | 0.003 | 0.074 | 0 | 0.001 | 0.004 | |
| bias Tauche | 0.001 | 0.001 | 0.067 | 0.001 | 0.001 | 0.006 | 0 | 0.012 | 0.007 | 0.004 | |




### 5.2.3. Ensemble of hourly precipitation fields on the Dudh Koshi River basin

Observed precipitation at measuring stations were then interpolated at the hourly time scale over the Dudh Koshi River basin at the 1-km spatial resolution. The method given by EQUATION 2 is applied, using shape parameters for the altitudinal factor selected TABLE 9. The average annual volumes of computed total precipitation ranged between 1365 mm and 1652 mm, and annual snowfall volumes ranged between 89 mm and 126 mm, in average over the 2014–2015 and 2015–2016 hydrological years. These values are consistent with other products available for the area. In particular, Savéan (2014) showed that the APHRODITE (Yatagai *et al.* 2012) product underestimates total precipitation over the Dudh Koshi River basin, with annual total precipitation of 1311 mm for the interannual average between 2001 and 2007, and Nepal *et al.* (2014) proposed a mean annual total precipitation for the Dudh Koshi basin of 2114 mm over the 1986–1997 period. The ERA-Interim reanalysis (25-km resolution) provided a mean annual precipitation of 1743 mm over the 2000–2013 period. Different relations between altitude and annual precipitation are then represented. The higher (lower) values are the positive (negative) rates, the sharpest are the spatial variations of annual precipitation. This has to be discussed considering the physical properties of convection at such high altitudes.

## 6. Conclusion

The main objective of this paper was to provide a representation of the effect of altitude on precipitation that represent spatial and temporal variability of precipitation in the Everest region. A weighted inverse distance method coupled with a multiplicative altitudinal factor was applied to spatially extrapolate measured precipitation to produce precipitation fields over the Dudh Koshi basin. The altitudinal factor for the Dudh Koshi basin is shown to acceptably fit a piecewise linear function of altitude, with significant seasonal variations. A sensitivity analysis was run to reduce the variation interval for parameters controlling the shape of the altitudinal factor. An uncertainty analysis was subsequently run to evaluated ensemble of simulated variables according to observed discharge for two small subcatchments of the Dudh Koshi basin located in mid- and high-altitude mountain environments.

Non deterministic annual water budgets are provided for two small gauged subcatchments located in high- and mid-altitude mountain environments. This work shows that the only uncertainties associated with representation of the orographic effect account for about 16% for annual total precipitation and up to 25% for simulated discharges. Annual evapotranspiration is shown to represent $26\% \pm 1\%$ of annual total precipitation for the mid-altitude catchment and $34\% \pm 3\%$ for the high-altitude catchment. Snow fall contribution is shown to be neglectible for the mid-altitude catchment and it represents up to $44\% \pm 8\%$ of total precipitation for the high-altitude catchment. These simulations at the local scale enhance current knowledge of the spatial variability of hydro-climatic processes in high- and mid-altitude mountain environments.

This work paves the way to produce hourly precipitation maps extrapolated from ground-based measurements that are reliable at the local scale. However, additional criteria would be needed to provide a single optimum parameter set for altitudinal



factor that would be suitable for the entire Dudh Koshi River basin. For exemple, snow cover areas simulated at a scale larger than the two catchments could be compared to available remote products (Behrangi *et al.* 2016). Independent measurements of precipitation could also be used to constrain the ensemble of precipitation fields.

Moreover, since observations are made over a very short duration and contain long periods with missing information, the results are limited to the 2014–2015 and 2015–2016 hydrological years and to the Dudh Koshi River basin. In addition, this study focuses only on one source of uncertainty in the measurement-spatialization-modeling chain, whereas sensitivity analysis should include all types of uncertainty (Beven 2015, Saltelli *et al.* 2006). A more complete method would include epistemic uncertainty on model parameters and aleatory uncertainty on input variables in the sensitivity analysis (Fuentes Andino *et al.* 2016).

## Acknowledgements

The authors extend special thanks to Professor Isabelle Sacareau (Passages Laboratory of the CNRS and Montaigne University of Bordeaux, France), coordinator of the Preshine Project. They are also grateful to the hydrometry team and the administrative staff of the Laboratoire Hydrosciences Montpellier, France, the hydrologists of the Institut des Geosciences de l'Environnement in Grenoble, France, the meteorologists of the Centre National de la Recherche Meteorologique in Toulouse and Grenoble, France, the Association Ev-K2 CNR and the Pyramid Laboratory staff in Bergamo, Italy, Kathmandu and Lobuche, Nepal, and the Vice-Chancellor of the Nepalese Academy of Science and Technology. The authors are particularly grateful to Professor Keith Beven from Lancaster University (UK) and to Dr. Patrick Wagnon from Institut de Recherche pour le Developpement. Finally, they pay homage to the local observers, local authorities, Sagarmatha National Park and the Cho-Oyu trekking agency with its respective staff and porters.

## Funding

This work was funded by the Agence Nationale de la Recherche (references ANR-09-CEP-0005-04/PAPRIKA and ANR-13-SENV-0005-03/PRESHINE), Paris, France. It was locally approved by the Bilateral Technical Committee of the Ev-K2-CNR Association (Italy) and the Nepal Academy of Science and Technology (NAST) within the Ev-K2-CNR/NAST Joint Research Project. It is supported by the Department of Hydrology and Meteorology, Kathmandu, Nepal.

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
