# Peer review of "Providing a non-deterministic representation of spatial variability of"

_Hydrology and Earth System Sciences, 2017_

## Referee Comment (RC1) · Anonymous Referee #1 · 9 Jun 2017

Review of

Eeckman, J., Chevallier, P., Boone, A., Neppel, L., De Rouw, A., Delclaux, F., and Koirala, D.: Providing a non-deterministic representation of spatial variability of precipitation in the Everest region, Hydrol. Earth Syst. Sci. Discuss., https://doi.org/10.5194/hess-2017-137, in review, 2017.

The authors address an important issue in analyzing precipitation with high impact on hydrological discharges in complex terrain, the effect of altitude on the spatial and temporal distribution of precipitation analysis in the Himalayan region. Non deterministic water budgets on an annual basis are given for two mountainous regions, indicating the

uncertainties associated with orographic effects for annual total precip and simulated run-off.

The paper is well written and clearly structured. Apart from some issues, which are listed below, the analysis and results are reasonable. Before publication, I would suggest to take into account the following comments:

1. The choice of the altitudinal beta-factors is crucial. How sensitive is the optimization to the choice of initial beta values? Some minimization algorithms suffer from inadequate initial values since they converge towards local minima which could be far away from the global ones. Did you make sensitivity studies with different initial values?

2. In 3.2, 3.2.1, it is stated that the precipitation analysis has been done by applying an IDW scheme. On which grid resolution is it done?

3. The ISBA surface scheme is used to simulate the exchange between atmosphere and soil/surface, using ECOCLIMAP surfaces. Why did the authors not use ECO-CLIMAP2 data? They should be higher resolved and available.

Some minor remarks and typos (but not all of them) are listed below:

P3L107: ". . . are presented in TABLE 2 . . ."

P3L108: "These time series present up to 61% missing values" sounds odd.

P4L121: replace "aleatory" by "random"

P7L201-202: The decrease of precipitation with altitude is characterized by various functions.

P8L229: "Optimal values that optimally fit . . ." sounds odd.

P12L344: "Based on values proposed in the literature, . . ."

P12L349-362: What are behavioral and nonbehavioral parameter sets? Please explain more detailed.

[Figure]

P13L373: The years 2013 and 2014 are used as spin-up period. Why such a long spin up time??

---

## Referee Comment (RC2) · Anonymous Referee #2 · 16 Jun 2017

The paper provides a non-deterministic representation of spatial variability of precipitation in the Everest region where is the mountainous area. Non deterministic annual water budgets are provided for two small catchments in mountainous area. It shows the uncertainties associated with the orographic effect for the main parameters of hydrology

The structure of the paper is good. However, I have few suggestions below to consider before proceed further.

Only one IDW is chosen for interpolation of precipitation. This method is or course simple and widely used for spatial interpolation of precipitation but this is deterministictype method which could not capture the effect of orographic in mountainous area. The authors should make a better review on different spatial interpolation methods for precipitation. They should try to look for such method that is called non-deterministic i.e. geostatistical methods...that are able to uncover the orographic effect.

Table 1 should be deleted; it is not cited in the text. It shows an overview of the observation network but information provided is not important. The location is already shown in the Figure 1 but should add some more information to that figure

P4, L121: I don't understand "Local measurements are necessarily biased by aleatory errors"

P4, L124 replace aleatory by random

P4, L131-134: difficult to understand, please rearrange your sentence again

I think that English should be improved. some sentences are unusual and difficult to understand

---

## Author Comment (AC1) · 21 Jun 2017

The responses are organized according to the organization of the questions. When necessary, the beginning of the question is repeated.

1. In order to answer this question in the paper, I will replace the text from P11L333 to P12L335 by the following text: The optimization method is highly sensitive to the choice of initial values for the beta1, beta2, z1 and z2 parameters. Several attempts have been done and the choices of initial ranges presented TABLE 6 are justified by the following arguments : - minimal and maximal values for the altitudinal thresholds z1 and z2 are chosen accordingly to both literature review (Barros et al. 2000,Anders

et al. 2006,Bookhagen and Burbank 2006, Shrestha et al. 2012, Nepal 2012, Savéan 2014) and observations. The first inquired altitudinal threshold is described in literature between 2000 m and 3000 m and the second threshold is described above 4000 m. These intervals have been enlarged to also test related values. - maximal (minimal) value for beta1 (beta2) are chosen about 10 times larger than the value computed based on observation. Considering the definition of the beta coefficient, a value of beta greater than 2km$\hat{}$-1 would lead to a multiplication of precipitation by 1.22 (by 0.82) within 100m. When applied to the precipitation observed at stations, this would lead to inconsistent precipitation when increasing altitude by 100m. - beta3 coefficient has to be negative because a positive value would lead to an unrealistic value at high summits. Moreover, the minimal value is chosen to be significantly smaller than the value computed for beta3 based on observations, but also to remain higher than the value computed for beta2 based on observations.

2. The IDW interpolation is performed over a 1 km resolution grid. This is specified P9L255.

3. The ECOCLIMAP2 product is only available for Europe (see Faroux et al., 2013). The studied area is not covered by ECOCLIMAP2. Consequently, I won't modify this point in the manuscript.

Minor remarks: I will correct the underlined minor remarksÂă: P3L107: these stations are presented TABLE 2 P7L201-202: The decreasing of precipitation with altitude is characterized P12L344: Based on values

In particular: - P4L121: replace "aleatory" by "random" I agree that the term 'random' would be more grammatically correct here. However, the term 'aleatory' is chosen in order to match the classification of uncertainties proposed by Beven,2016. In Beven,2016, aleatory error is defined as 'uncertainty with stationary statistical characteristics. May be structured (bias, autocorrelation, long term persistence) but can be reduced to a stationary random distribution'. Consequently, I won't modify this point in

the manuscript.

- P12L349-362: What are behavioral and nonbehavioral parameter sets? Please explain more detailed.

Behavioral parameter sets are defined P9L248: A behavioral parameter set is a set that respects conditions (maximum or minimum thresholds) on the output of the orographic precipitation model. At P12L338, I will insert the following sentence : A behavioral parameter set is a set that leads to an annual amount of total precipiation for both catchments comprised between the minimal and maximal values presented Table7. A parameter sets that does not meet these conditions is considered as non-behavioral.

- Discussion paperP13L373: The years 2013 and 2014 are used as spin-up period. Why such a long spin-up time?? The hydrological year that runs from April,1st 2013 to March, 31rd 2014 (i.e. 365 days) is used as spin-up period. 365 days are considered to be a necessary spin-up period to set up all reservoirs at a representative volume, in particular for snow pack and soil water storage. P13L373 , I will add: The 2013–2014 hydrological year was used [...]

---

## Author Comment (AC2) · 21 Jun 2017

The responses are organized according to the organization of the questions. Extracts of the questions are repeated and the answers are introduced by (*).

Only one IDW is chosen for interpolation of precipitation[...]. They should try to look for such method that is called non-deterministic i.e. geostatistical methods.

* An important review of interpolation method suitable for mountainous areas has been done. In particular, three of the co-authors have applied the cokriging geostatistical interpolation method to estimate the monsoon precipitation in the Koshi River basin

(Gonga et al., 2016) . The IDW coupled with a multiplicative altitudinal factor (Valery et al, 2010) has been chosen in this paper because it presents the advantage to separate the effect of (x,y) position from the effect of altitude. The effect of altitude can therefore be independently studied and the controlling parameters have physical meaning. Values can then be provided for altitudinal lapse rates and thresholds for precipitation.

In order to answer this question in the paper, I will add the following text P7L192: This method presents the advantage to separate the effect of (x,y) position from the effect of altitude. The effect of altitude can therefore be independently studied and the controlling parameters have physical meaning.

Table 1 should be deleted; it is not cited in the text:

* I will add reference to Table 1 in the main text.

P4, L121: I don't understand "Local measurements are necessarily biased by aleatory errors".

* For a better understanding, I will reformulate this sentence in the text asÂă: 'Local measurements can not be an exact quantification of any climatic variables, and they are necessarily associated by errors that follow an random distribution law. ' A large number of factor can indeed affect local measurements of climatic variables (e.g. approximation in the sensor records, influence of variations faster that the time step, local site effects, . . .).

P4, L124 replace aleatory by random

*I agree that the term 'random' would be more grammatically correct here. However, the term 'aleatory' is chosen in order to match the classification of uncertainties proposed by Beven,2016. In Beven,2016, aleatory error is defined as 'uncertainty with stationary statistical characteristics. May be structured (bias, autocorrelation, long term persistence) but can be reduced to a stationary random distribution'. Consequently, I won't modify this point in the manuscript.

P4, L131-134: difficult to understand, please rearrange your sentence again

* I will replace this sentence by: Water level time series are available from March 2014 to March 2015. Time serie at Kharikhola station contains 34% of missing data in 2014-2015, corresponding to damages to the sensor (TABLE 3).

―――――――――――――

---

## Author Response (AR1)

**Author's response to Review1**

The responses are organized according to the organization of the questions. When necessary, the beginning of the question is repeated.

- *The choice of the altitudinal beta-factors is crucial. How sensitive is the optimization to the choice of initial beta values?[...]*
  In order to answer this question in the paper, the text from P11L333 to P12L335 is remplaced by the following text :
  The optimization method is highly sensitive to the choice of inital values for the $\beta_1$, $\beta_2$, $\beta_3$, z1 and z2 parameters. Several attempts have been done and the choices presented TABLE 6 are justified by the following arguments :
  - Minimum and maximum values for the altitude thresholds z1 and z2 are chosen accordingly to both litterature review (Barros et al., 2000; Anders et al., 2006; Bookhagen and Burbank, 2006; Shrestha et al., 2012; Nepal, 2012; Savéan, 2014) and observations. The first inquired altitudinal threshold is described in litterature between 2000 m and 3000 m and the second threshold is described above 4000 m. These intervals have been enlarged to also test related values.
  - Maximum (resp. minimum) value for $\beta_1$ (resp. $\beta_2$) are chosen about 10 times larger than the value computed based on observation. Considering the definition of the beta coefficient, a value greater than 2 $km^{-1}$ (resp. lower than -2 $km^{-1}$) would lead to a multiplication of precipitation by 1.22 (resp. by 0.82) within 100 m. When applied to the precipitation oberved at stations, this would lead to inconsistent precipitation when increasing altitude by 100 m.
  - The $\beta_3$ coefficient has to be negative because a positive value would lead to unrealistic values at high altitudes. Moreover, the minimum value is chosen to be significantly smaller than the value computed for $\beta_3$ based on the observations, but also to remain higer than the value computed for $\beta_2$ based on the observations.

- *it is stated that the precipitation analysis has been done by applying an IDW scheme. On which grid resolution is it done?*
  The IDW interpolation is performed over a 1 km resolution grid. This is specified P9L255. Consequently, the manuscript is not modified concerning this point.

- *[...]Why did the authors not use ECO-CLIMAP2 data?*
  The ECOCLIMAP2 product is only available for Europe (see (Faroux et al., 2013)). The studied area is not covered by ECOCLIMAP2. The following sentence is inserted P11L352 :
  The ECOCLIMAP2 product (Faroux et al., 2013) is derived from ECOCLIMAP1 and provides enhanced descriptions of surfaces. However, ECOCLIMAP2 is only available over Europe and therefore is not used in this study.

- *Minor remarks*
  The minor remarks are corrected in the manuscript :
  - P3L107 : these stations are presented TABLE 2
  - P7L201-202 : The descreasing of precipitation with altitude is characterized
  - P12L344 : Based on values
  - *P4L121 : replace "aleatory" by "random"*
    I agree that the term 'random' would be more grammatically correct here. However, the term 'aleatory' is chosen in order to match the classification of uncertainties proposed by Beven,2016. In (Beven, 2015), aleatory error is defined as 'uncertainty with stationary statistical characteristics. May be structured (bias,autocorrelation, long term persistence) but can be reduced to a stationary random distribution'. Consequenly, I won't modify this point in the manuscript.

- *P12L349-362 : What are behavioral and nonbehavioral parameter sets ? Please explain more detailed.*

Behavioral parameter sets are defined P9L248 : A behavioral parameter set is a set that respects conditions (maximum or minimum thresholds) on the output of the orographic precipitation model. At P12L338, the following sentence is inserted in the manuscript : A behavioral parameter set is a set that leads to an annual amount of total precipiation for both catchments comprised between the minimal and maximal values presented Table7. A parameter sets that does not meet these conditions is considered as non-behavioral.

- *The years 2013 and 2014 are used as spin-up period. Why such a long spin-up time ? ?*
The hydrological year that runs from April,1st 2013 to March, 31rd 2014 (i.e. 365 days) is used as spin-up period. 365 days are considered to be a necessary spin-up period to set up all reservoirs at a representative volume, in particular for snow pack and soil water storage. P13L373 , the following sentence is added : The 2013–2014 hydrological year was used [...]

**Author's response to Review2**

The responses are organized according to the organization of the questions. Extracts of the questions are repeated and the answers are introduced by (*).

- *Only one IDW is chosen for interpolation of precipitation[. . .]. They should try to look for such method that is called non-deterministic i.e. geostatistical methods.*

An important review of interpolation method suitable for mountainous areas has been done. In particular, three of the co-authors have applied the cokriging geostatistical interpolation method to estimate the monsoon precipitation in the Koshi River basin (**?**) . In order to answer this question in the paper, the following text is inserted P7L192 :
However, few studies provide precipitation fields at the hourly time scale (**??**) and precipitation fields at spatial scales lower than 1 km are always obtained using altitude linear lapse rates (**???**). However, the considered lapse rates are constant in time and/or uniform in space. The spatial and temporal variability of the precipitation is then not represented in these studies. Moreover, the geostatistical cokriging method has been applied by **?** for monsoon precipitation interpolation over the Koshi catchment. However, the provided precipitation fields overally underestimate the observations, and this method is shown not to be adequate for the interpolation of solid precipitation.

and the following text is inserted P7L192 :
This method presents the advantage of using an altitudinal factor which can vary in time and space. The spatial and temporal variability of the precipitation is therefore represented in this method. Moreover, the effect of altitude is independently studied and the controlling parameters have physical meaning.

- *Table 1 should be deleted ; it is not cited in the text*
The reference to TABLE 1 is added P5L106.

- *P4L121 : I don't understand "Local measurements are necessarily biased by aleatory errors".*
For a better understanding, this sentence is reformulated as : 'Local measurements can not be an exact quantification of any climatic variables, and they are necessarily associated by errors that follow an random distribution law. ' A large number of factor can indeed affect local measurements of climatic variables (e.g. approximation in the sensor records, influence of variations faster that the time step, local site effects, . . . ).

- *P4, L124 replace aleatory by random*
  I agree that the term 'random' would be more grammatically correct here. However, the term 'aleatory' is chosen in order to match the classification of uncertainties proposed by Beven,2016. In Beven,2016, aleatory error is defined as 'uncertainty with stationary statistical characteristics. May be structured (bias,autocorrelation, long term persistence) but can be reduced to a stationary random distribution'. Consequenly, I won't modify this point in the manuscript.

- *P4, L131-134 : difficult to understand, please rearrange your sentence again*
  This sentence is modified as :

[revised manuscript text omitted]